# Folic Acid Ameliorates Renal Injury in Experimental Obstructive Nephropathy: Role of Glycine N-Methyltransferase

**DOI:** 10.3390/ijms24076859

**Published:** 2023-04-06

**Authors:** Ko-Lin Kuo, Chin-Wei Chiang, Yi-Ming Arthur Chen, Chih-Chin Yu, Tzong-Shyuan Lee

**Affiliations:** 1Division of Nephrology, Taipei Tzu Chi Hospital, Buddhist Tzu Chi Medical Foundation, New Taipei 231405, Taiwan; 2School of Medicine, Buddhist Tzu Chi University, Hualien 97004, Taiwan; 3School of Post-Baccalaureate Chinese Medicine, Tzu Chi University, Hualien 97004, Taiwan; 4Department of Physiology, National Yang Ming Chiao Tung University, Taipei 112304, Taiwan; k366988@gmail.com; 5Graduate Institute of Biomedical and Pharmaceutical Science, Fu Jen Catholic University, New Taipei 24205, Taiwan; 150110@mail.fju.edu.tw; 6Division of Urology, Department of Surgery, Taipei Tzu Chi Hospital, Buddhist Tzu Chi Medical Foundation, New Taipei 231405, Taiwan; b91401049@gmail.com; 7College of Medicine, Tzu Chi University, Hualien 97004, Taiwan; 8Graduate Institute, Department of Physiology, College of Medicine, National Taiwan University, Taipei 100, Taiwan

**Keywords:** folic acid, glycine N-methyltransferase, unilateral ureteral obstruction

## Abstract

Folic acid exerts both anti-inflammatory and antifibrotic effects. Glycine N-methyltransferase (GNMT), the major folic acid-binding protein in the liver, is a crucial enzyme that regulates the cellular methylation process by maintaining S-adenosylmethionine levels. However, as yet neither the therapeutic effects of folic acid in renal fibrosis nor whether GNMT is involved in these folic acid-associated mechanisms has been investigated. First, the expression of GNMT was examined in human kidneys with or without obstructive nephropathy. Later, wild-type and *GNMT* knockout (*GNMT^−/−^*) mice were subjected to unilateral ureteral obstruction (UUO) and then treated with either folic acid or vehicle for 14 days. Renal tubular injury, inflammation, fibrosis, and autophagy were evaluated by histological analysis and Western blotting. We observed increased expression of GNMT in humans with obstructive nephropathy. Furthermore, UUO significantly increased the expression of GNMT in mice; in addition, it caused renal injury as well as the development of both hydronephrosis and tubular injury. These were all alleviated by folic acid treatment. In contrast, *GNMT^−/−^* mice exhibited exacerbated UUO-induced renal injury, but the protective effect of folic acid was not observed in *GNMT^−/−^* mice. We propose a novel role for folic acid in the treatment of renal fibrosis, which indicates that GNMT may be a therapeutic target.

## 1. Introduction

Chronic kidney disease (CKD) is an emerging health problem that imposes a growing socioeconomic burden on societies worldwide [1,2,3,4,5]. A common pathologic feature of CKD is inflammatory cell infiltration occurring in the injured kidneys in early stages, followed by tubulointerstitial fibrosis in the later stages of disease progression [2,3,5]. Among patients with CKD, upper urinary tract obstruction that results in renal dysfunction is a significant clinical problem in both adult and pediatric populations. The progression of renal dysfunction in obstructive nephropathy is associated with both acute renal tubular injury and inflammatory responses. In addition, chronic obstructive nephropathy induces the activation of resident fibroblasts, which leads to renal interstitial fibrosis and glomerulosclerosis [6,7].

Folic acid (pteroylmonoglutamate) is a water-soluble form of vitamin B_9_, which can be obtained from natural foods such as beans, liver, and dark green leafy vegetables. Folic acid plays an important role in maintaining the normal functions of cells. It acts as a cofactor in DNA synthase and cellular methylation processes. Folic acid status has been investigated in association with the development of anemia, neuronal tube defects, and carcinogenesis [8]. Clinically, folic acid supplementation has been shown to ameliorate the risk of cardiovascular disease in patients with end-stage renal disease [9,10,11]. In addition, treatment with folic acid increased DNA methylation and reduced inflammation in gastric cancer [12]. Several lines of evidence have confirmed the beneficial anti-inflammatory and antifibrotic effects of folic acid in certain disease models. Daily supplementation with folic acid in drinking water inhibited pulmonary artery obstruction-induced right ventricular failure by diminishing collagen deposition, autophagy induction, and reactive oxygen species (ROS) production in mice [13]. Additionally, treatment with folic acid restored CCl_4_-induced hepatic injury by reducing both the level of oxidative stress and the inflammatory response [14]. Intracerebroventricular injection of folic acid was found to exert an antidepressant effect in a forced swimming and tail suspension-induced depression model [15].

Glycine N-methyltransferase (GNMT), a methyltransferase, is expressed in various tissues, including the liver, pancreas, jejunum, submaxillary gland, and kidney [16]. The GNMT complex consists of four identical subunits with 292 amino acid residues in each subunit. The N-terminus of GNMT interacts with other subunits and regulates enzyme activity. The major function of GNMT involves regulating the cellular methylation process by first catalyzing SAM and then forming SAH and N-methylglycine [17]. In addition to its catalytic activity, GNMT also exhibits anticarcinogenic activity due to its antiproliferative effect upon translocation into the nucleus [18]. It has also been proposed that GNMT participates in regulating gluconeogenesis [16,19]. A deficiency of GNMT has been reported to promote the development of hepatocellular carcinoma in both humans and mice [20,21]. Our previous study revealed that deletion of GNMT exacerbates the consequences of atherosclerosis and ulcerative colitis [22,23], implying that GNMT expression is important in inflammatory diseases. In addition, GNMT functions as the major folic acid-binding protein, and its enzyme activity is inhibited by binding with folic acid derivatives [24,25]; however, the role of GNMT in folic acid-mediated functions remains to be elucidated.

In the present study, we investigated the potential role and underlying mechanism of folic acid and its major binding protein, GNMT, in the pathogenesis of obstructive nephropathy via a unilateral ureteral obstruction (UUO) mouse model; this is a well-known model of obstructive nephropathy associated with renal fibrosis [26,27]. First, we examined the expression of GNMT not only in kidney specimens from patients with hydronephrosis but also in UUO mice. Second, we assessed the protective effect of folic acid against UUO-induced inflammation and interstitial fibrosis in the kidneys of wild-type (WT) mice and knockout mice with genetic deletion of *GNMT*.

## 2. Results

### 2.1. Increased GNMT Expression in Humans and Mice with Obstructive Nephropathy

To validate the clinical significance and pathogenic role of GNMT in human noninfectious nephritis, nephrectomized specimens of affected kidneys were obtained from patients with unilateral ureteral obstruction (*n* = 6). The causes of obstruction (also called hydronephrosis) were urological cancer and kidney stones. Normal tissues from the nephrectomized kidneys of renal cell carcinoma patients were used as controls (*n* = 3). We found that the expression of GNMT in immunohistochemical staining was increased in the hydronephrosis group compared with the control group (Figure 1). Moreover, 3, 7, and 14 days after UUO surgery, the protein expression of GNMT was upregulated in a time-dependent manner in WT mice (Figure 2A). We also used immunofluorescence staining to evaluate the expression of GNMT. Immunoblotting showed that GNMT was expressed in renal tubular cells in the sham group; however, the level of GNMT was increased in the interstitium 14 days after UUO surgery (Figure 2B). Thus, these data suggest that GNMT expression may be associated with the progression of obstructive nephropathy.

### 2.2. Treatment with Folic Acid Reduces Renal Injury in UUO Mice but Not in GNMT^−/−^ Mice

We further sought to delineate the possible role of GNMT in folic acid supplementation therapy. Eight-week-old *GNMT^−/−^* mice underwent UUO surgery and then received folic acid treatment for 14 days. We found that treatment with folic acid decreased UUO-induced hydronephrosis in WT mice, while deletion of GNMT exacerbated UUO-induced hydronephrosis. Administration of folic acid did not affect hydronephrosis in *GNMT^−/−^* mice (Figure 3). Based on histological examination, we found that UUO induced an increase in renal tubular injury, with a subsequent increase in leukocyte infiltration (Figure 4A–C). Furthermore, Masson’s trichrome staining revealed that UUO induced the deposition of collagen in the interstitial space (Figure 4D,E). The deletion of GNMT exacerbated these UUO-induced histological changes. The UUO-induced increases in renal tubular injury, leukocyte infiltration, and interstitial fibrosis were ameliorated by treatment with folic acid; however, the protective effects of folic acid were not observed in *GNMT^−/−^* mice (Figure 4). The degree of renal injury was evaluated, and the disease activity index (Figure 4F) indicated that UUO induced an increase in renal injury; moreover, this increase was aggravated in *GNMT^−/−^* mice. Treatment with folic acid reduced renal injury in WT mice but not in *GNMT^−/−^* mice (Figure 4F). Thus, these data suggest that treatment with folic acid mitigates UUO-induced renal injury in a GNMT-dependent manner.

### 2.3. Treatment with Folic Acid Inhibits the Inflammatory Response and Decreases the Expression of Fibrosis-Related Proteins in UUO Mice but Not in GNMT^−/−^ Mice

In response to renal tubular injury, microvascular endothelial cells modify intercellular junctions and express adhesion molecules to recruit circulating leukocytes to the site of tissue injury. The infiltration of leukocytes subsequently eliminates injured tissue and repairs damaged tissue [28]. Therefore, we examined the role of folic acid and GNMT in the UUO-induced renal inflammatory response. Our data showed that UUO induced the expression of both intercellular adhesion molecule 1 (ICAM-1) and vascular cell adhesion molecule 1 (VCAM-1) in WT mice. Deletion of GNMT exacerbated changes in the microvascular architecture. Treatment with folic acid reduced the levels of adhesion molecules in WT mice but not in *GNMT^−/−^* mice (Figure 5A,B). The inflammatory marker inducible nitric oxide synthase (iNOS) was also higher in *GNMT^−/−^* mice when compared with WT mice (Figure 5A,B).

Changes in the microvascular structure and expression of adhesion molecules recruit circulating leukocytes, which undergo transmigration to the site of tissue injury. Our data showed that the UUO-induced infiltration of leukocytes, including macrophages, T cells, and neutrophils, was significantly reduced after treatment with folic acid (Figure 5C,D). Conversely, the deletion of GNMT aggravated UUO-induced leukocyte infiltration, and treatment with folic acid had no effect on leukocyte infiltration in *GNMT^−/−^* mice (Figure 5C,D).

Because our data indicated that UUO induced both collagen deposition and interstitial fibrosis (Figure 4D,E), we further examined the expression of fibrosis-related proteins. An increase in the level of transforming growth factor-β (TGF-β) stimulates fibroblasts to synthesize stress fibers, express α-smooth muscle actin (α-SMA), and differentiate into myofibroblasts. Myofibroblasts further produce collagen and result in renal fibrosis [7,29]. Our data showed that UUO induced TGF-β expression (Figure 6). Furthermore, α-SMA expression was increased after UUO surgery (Figure 6). Collagen deposition and thickening of the basement membrane were identified; UUO induced both collagen deposition and interstitial fibrosis with increased levels of collagen type I alpha-1 chain (COL1A1) and type IV alpha-2 chain (COL4A2). The expression of fibrosis-related proteins was markedly increased in *GNMT^−/−^* mice compared with WT mice (Figure 6A,B). Treatment with folic acid decreased fibrosis-related expression and diminished collagen deposition in WT mice; however, this effect was not observed in *GNMT^−/−^* mice (Figure 6A,B).

### 2.4. Treatment with Folic Acid Activates Autophagy Flux in UUO Mice but Not in GNMT^−/−^ Mice

Physiologically, autophagy is considered a protective process against environmental stress, such as nutrition deprivation [30,31]. Autophagy also plays a potential role in regulating both the inflammatory response and fibroblast activation [32,33]. In addition, autophagy is associated with the development of renal tubular injury [34]. Therefore, we examined whether autophagy was involved in the protective effect of folic acid in UUO mice. Our data indicated that UUO induced an increase in levels of LC3-II and p62, suggesting that autophagy flux was impaired in the pathogenesis of UUO-induced renal injury (Figure 7A–C). That treatment with folic acid rescued such an impairment of autophagy flux was evident in the fact that folic acid increased the levels of LC3-II and decreased the levels of p62 in UUO mice (Figure 7A–C). However, this effect of folic acid on autophagy activation was abolished in *GNMT^−/−^* mice (Figure 7A–C). Additionally, the activity of protein kinases, including Akt and p70s6k, which is related to autophagy activation [35], increased; however, these increases diminished in UUO WT mice after 14 days of folic acid treatment (Figure 7D–G). Likewise, treatment with folic acid decreased the UUO-induced increase in mTOR activity (Figure 7D–G), which has an inhibitory effect on autophagy activation [35]. Moreover, folic acid failed to induce the UUO-mediated alterations in Akt, p70s6k and mTOR in *GNMT^−/−^* mice (Figure 7D–G). Thus, these data suggest that the therapeutic effect of folic acid may be attributed to the activation of autophagy flux in a GNMT-dependent manner; in this way, it slows down the progression of renal injury. A schematic illustration of the proposed mechanism by which folic acid protects against UUO-induced renal injury via GNMT and the autophagy signaling pathway in UUO mice is shown in Figure 8.

## 3. Discussion

Although the biological functions of folic acid have been well established, the underlying molecular mechanisms of its protective effect in renal fibrosis have not been reported. GNMT has been recognized as a folic acid-binding protein [25]; however, the exact role of GNMT in folic acid-mediated functions is largely unknown. Our human data first demonstrated that the expression of GNMT is increased in subjects with hydronephrosis, and an in vivo study also showed increased expression of GNMT after UUO surgery. Later, we found that folic acid treatment dramatically mitigated UUO-induced hydronephrosis, renal tubular injury, inflammation, and fibrosis by modulating autophagy in mice. However, the therapeutic effect of folic acid was not observed in *GNMT^−/−^* mice, which suggests that the protective effects of folic acid were mediated by GNMT. Moreover, GNMT deficiency aggravated obstructive nephropathy by increasing autophagy, which in turn resulted in increased renal tubular injury. Subsequently, the degrees of inflammation and fibrosis were increased in *GNMT^−/−^* mice. To the best of our knowledge, this is the first study to demonstrate that GNMT plays a vital role in the development of obstructive nephropathy.

GNMT has also exhibited an antiproliferative effect in cancer cells [18], and it is characterized as a key protein in modulating the SAM/SAH ratio to prevent aberrant methylation. Activation of FOXO-GNMT increases the level of SAM to prevent energy waste and protect against severe inflammation [36]. These findings imply that the expression of an adequate level of GNMT may provide a protective effect against cell damage. Our previous studies demonstrated that the expression of GNMT was increased in atherosclerosis and dextran sulfate sodium (DSS)-induced colitis. Moreover, during the inflammatory response the expression of GNMT seems to be restricted to the infiltrated leukocytes, including neutrophils, T cells, and macrophages. Deletion of GNMT exacerbated the inflammatory response in atherosclerosis and DSS-induced colitis [22,23]. In addition, the absence of GNMT increased natural killer cell activation and cytotoxicity [37]. These findings suggest that GNMT may play an important role in regulating inflammation. We found that inflammation markers were consistently higher in *GNMT^−/−^* mice than in WT mice, and that leukocyte infiltration was exacerbated in *GNMT^−/−^* mice. Our data revealed that the deletion of GNMT exacerbates renal tubular injury, which further supports this hypothesis.

Several clinical and experimental studies have revealed that treatment with folic acid exerts a protective effect in different types of inflammatory diseases. A meta-analysis suggested that folic acid therapy reduces the risk of cardiovascular disease in patients with end-stage renal disease [9]. Moreover, in a mouse model of *Helicobacter pylori*-associated gastric cancer, feeding a folic acid-supplemented diet ameliorated the inflammatory response by altering inflammation-associated gene expression [12]. In addition, the administration of folic acid reduced lipopolysaccharide-induced intrauterine growth restriction by decreasing inflammatory cytokine production in pregnant mice [38]. It has been demonstrated that treatment with folic acid restores endothelial nitric oxide synthase coupling, subsequently preventing vascular remodeling and decreasing superoxide production in angiotensin II-infused hyperphenylalaninemia mice [39]. Consistent with these observations, our data demonstrated an anti-inflammatory effect of folic acid via a decrease in inflammatory cell infiltration. In addition, treatment with folic acid preserved the microvascular architecture, as shown by reduced levels of adhesion molecules and inflammation markers.

Prolonging the inflammatory response triggers tissue repair mechanisms, including the activation of myofibroblasts and the production of extracellular matrix, which results in tissue fibrosis. In response to injury, activation of the wound healing process restores the architecture and function of the kidney. However, overactivation of the repair mechanism leads to renal fibrosis, which can be defined as tubulointerstitial fibrosis and glomerulosclerosis, and decreases renal function [40]. Renal fibrosis leads to a decline in renal function and further causes end-stage renal disease. Folic acid treatment reduces tissue fibrosis by lessening oxidative stress in different rodent models. For instance, folic acid ameliorated CCL_4_-induced liver fibrosis by reducing inflammation and RO production [14]. In addition, folic acid reduced angiotensin II-induced glomerulosclerosis by decreasing the expression of type IV collagen and matrix metalloproteinases [41]. Hyperhomocysteinemia-induced glomerular injury was decreased by reducing the expression of TGF-β, α-SMA, and nephrin loss after folic acid treatment [42]. According to the Renal Substudy of the China Stroke Primary Prevention Trial (CSPPT), folic acid therapy reduced the risk of CKD progression and the estimated glomerular filtration rate decline. CKD patients benefited more from this therapy, with reductions of 56% and 44% in the risk for progression of CKD and the rate of eGFR decline, respectively [10,11]. The clinical results of CSPPT are mutually supported by our present in vivo study showing that treatment with folic acid reduced both renal leukocyte infiltration and interstitial fibrosis.

Mechanical stretching has been shown to trigger autophagy [43]. After urinary tract obstruction, the increase in intratubular pressure may induce autophagy. Excessive autophagy has been shown to promote renal tubular atrophy and apoptotic cell death in obstructed tubular cells [44]. In addition, autophagy activation is associated with the promotion of inflammation [33]. However, autophagy activation is essential for the activation of fibroblasts [32]. Several studies have demonstrated that *GNMT^−/−^* mice display abnormal activation of signaling pathways, including those of Ras and Jak/STAT, in hepatocarcinoma [21,45], which may reflect a high level of autophagy [46,47]. In addition, the overexpression of GNMT increases mTOR activity via binding to DEP domains containing mTOR-interacting protein (DEPTOR), an endogenous mTOR inhibitor [48]. These studies suggest that GNMT may be involved in regulating autophagy. In line with this hypothesis, our data indicated that the deletion of GNMT aggravated UUO-induced autophagy, decreased mTOR activation, and increased LC3-II levels. Furthermore, our data demonstrated that treatment with folic acid decreased autophagy formation. This finding is consistent with the observation that the administration of folic acid decreased both autophagy induction and ROS production and consequently reduced right ventricular failure in a murine model [13]. In addition, it has been demonstrated that cellular methionine/SAM levels modulate autophagy by regulating the methylation of PP2A, which negatively regulates mTOR activity [49]. 

There are several important aspects of this study that merit discussion. First, our investigation of the novel role of folic acid in treating renal fibrosis, a significant worldwide health issue, is a major strength of our research. Second, this study highlights the involvement of GNMT in the folic acid-related mechanisms relevant to renal fibrosis. Given this, it is crucial to identify potential GNMT-inducers for prevention or treatment of renal fibrosis in the future. Third, the study’s use of both human kidney samples and animal models adds to the validity of the findings. Fourth, our evaluation of multiple outcomes, including renal tubular injury, inflammation, fibrosis, and autophagy, provides a comprehensive evaluation of folic acid’s effects on renal fibrosis. However, it is important to note that our study also has limitations. Firstly, as animal models were used, the findings may not directly translate to human physiology. Secondly, while the study sheds light on the effects of folic acid and GNMT on renal fibrosis, it does not suggest a detailed mechanism for how they interact to produce these effects.

## 4. Materials and Methods

### 4.1. Reagents and Antibodies

The following antibodies were obtained from Santa Cruz Biotechnology (Santa Cruz, CA, USA): goat anti-COL1A1, anti-p-p70s6k (Thr389), and anti-Akt; mouse anti-α-SMA, anti-LC-3, anti-mTOR, and anti-GNMT; rat anti-CD3; and rabbit anti-p70s6k, anti-p62, anti-p-mTOR (Ser2448), and anti-COL4A2. Mouse anti-p-Akt (Ser473) and rabbit ant-iNOS and anti-MPO antibodies were obtained from Cell Signaling Technology (Beverly, MA, USA). Rat anti-F4/80 and rabbit anti-ICAM-1 and anti-VCAM-1 antibodies were obtained from Abcam (Cambridge, MA, USA). Both the mouse anti-GAPDH antibody and Masson’s trichrome staining kit were obtained from Sigma–Aldrich (St. Louis, MO, USA). The mouse anti-TGF-β antibody was obtained from R&D Systems (Minneapolis, MN, USA).

### 4.2. Human Kidney Specimens and Experimental Animals

Nephrectomy specimens of affected human kidneys were obtained from patients with unilateral ureteral obstruction; obstructions were caused by urological cancer and kidney stones. Normal tissues from the nephrectomized kidneys of renal cell carcinoma patients were used as controls. This part of the study was approved by the Institutional Review Boards of Taipei Tzu Chi Hospital (IRB approval number: 09-XD-010), and informed consent was obtained from all participants. In addition, all experimental procedures and protocols involving animals were carried out in accordance with the institutional animal care committee of Taipei Tzu Chi Hospital and complied with the Guide for the Care and Use of Laboratory Animals (IACUC approval number: 108-IACUC-003). Male wild-type (WT) C57BL/6 mice were obtained from the National Laboratory Animal Center, National Science Council (Taipei, Taiwan); *GNMT*^−/−^ mice on a C57BL/6 background were obtained from the Yi-Ming Arthur Chen laboratory (Taipei, Taiwan). Mice were housed in barrier facilities on a 12-h light/12-h dark cycle and fed a normal chow diet.

### 4.3. Unilateral Ureteral Obstruction (UUO) and Folic Acid Treatment

A total of 32 mice (WT: *n* = 16, *GNMT^−/−^*: *n* = 16) were randomly divided into two groups: the control group and the folic acid treatment group. Eight- to ten-week-old male WT and *GNMT^−/−^* C57BL/6 mice were anesthetized with pentobarbital (80 mg/kg, intraperitoneal injection), and then the left kidney was exposed through a dorsal midline skin incision. Next, the ureter was ligated with two independent 4-0 nylon sutures. Lastly, the muscle and skin were closed with 5-0 Ethilon sutures. Mice in the sham group were subjected to the same operation but without ureteral ligation. Mice subjected to UUO surgery received daily treatment with folic acid (4 mg/kg) by intraperitoneal injection. Kidneys from both sham mice and UUO mice were harvested at the indicated times after ureteral ligation.

### 4.4. Immunohistochemical Analysis

Kidney sections were deparaffinized, rehydrated, and incubated with 3% H_2_O_2_ for 10 min. After blocking with 1% BSA for 1 h, the samples were incubated with an anti-GNMT antibody overnight at 4 °C and then incubated with a fluorescein isothiocyanate-conjugated or HRP (horseradish peroxidase)-conjugated secondary antibody for 2 h at 37 °C. Immunoreactions were visualized under a TE2000-U fluorescence microscope (Nikon, Melville, NY, USA).

### 4.5. Western Blot Analysis

Tissues were lysed with phosphate-buffered saline containing 1% Triton X-100, 0.1% SDS, 0.5% sodium deoxycholate, 1 μg/mL leupeptin, 10 μg/mL aprotinin, 1 mM PMSF, Tyr phosphatase cocktail I, and Ser/Thr phosphatase cocktail II on ice. After sonication, tissue extracts underwent centrifugation at 12,000× *g* for 5 min at 4 °C. The supernatants were collected as tissue lysates. All protein concentrations were examined using the Bradford protein–binding assay. Aliquots (50 μg of protein) of lysates were separated on either 8% or 10% SDS–PAGE and then transblotted onto an Immobilon^TM^-P membrane (Millipore, Bedford, MA, USA). After blocking with 5% skim milk for 1 h, the membranes were incubated first with primary antibodies and then with secondary antibodies. Protein bands were detected using an enhanced chemiluminescence kit (PerkinElmer, Boston, MA, USA) and quantified with ImageQuant 5.2 software (Health care Bio-Sciences, Lancaster, PA, USA).

### 4.6. Histological Examination

Harvested kidneys were fixed with 4% paraformaldehyde for two days and then dehydrated and embedded in paraffin. Tissue blocks were cut into 8 μm sections. Kidney section slides were rehydrated and then underwent hematoxylin and eosin (H&E) staining. The stained sections were viewed under a Motic Type 102M microscope. The degree of renal tubular injury, infiltration of leukocytes, and area of fibrosis were quantified according to previous methods [26] with minor modifications, as shown in Table 1. The disease activity index was determined and renal injury was assessed.

### 4.7. Masson’s Trichrome Staining

Kidney sections were deparaffinized, rehydrated, and fixed with Bouin’s solution at 65 °C for 15 min. The sections were washed with PBS until the yellow color disappeared. After washing with PBS, nuclei were stained with Weigert’s iron hematoxylin for 5 min and again washed with PBS. The cytosol was stained with Biebrich scarlet-acid fuchsin solution for 15 s. Next, the sections were incubated with phosphomolybdic/phosphotungstic acid solution for 15 min, and then collagen was stained with aniline blue solution for 15 min. Slides were dehydrated and mounted, and the fibrotic area was viewed under a Motic Type 102M microscope.

### 4.8. Statistical Analysis

All data are presented as means ± SEM values from ten independent experiments. The Mann–Whitney test was used for comparisons between two independent groups. The Kruskal–Wallis test, followed by the Bonferroni post hoc test, was used for comparisons among multiple groups. Differences where *p* < 0.05 were considered statistically significant.

## 5. Conclusions

In conclusion, we elucidated a possible underlying molecular mechanism of folic acid in preventing kidney fibrosis after ureter obstruction. Given previous studies from other groups as well as our own studies, we hypothesize that the deletion of GNMT results in both an abnormal SAM/SAH ratio and the overactivation of several signaling pathways, including autophagy; this results in aberrant methylation of DNA and increases both the inflammatory response and fibrosis. This is the first time that we have identified the role of GNMT in folic acid supplementation therapy. Our findings reveal that treatment with folic acid produces anti-inflammatory and antifibrotic effects in experimental obstructive nephropathy by decreasing autophagy in a GNMT-dependent manner. In addition, deletion of GNMT exacerbates UUO-induced renal injury. Therefore, we conclude that GNMT may be a therapeutic target for renal fibrosis, and that folic acid may be used as a treatment. Because of its potential therapeutic benefits in preventing CKD progression, folic acid therapy should be considered for the clinical management of CKD.

## Figures and Tables

**Figure 1 ijms-24-06859-f001:**
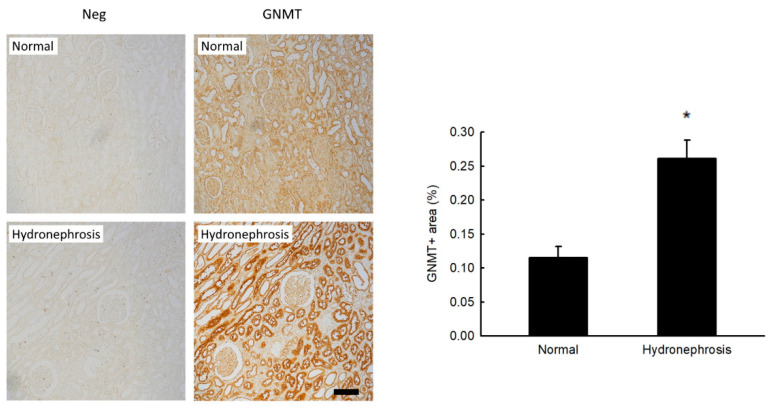
GNMT expression is upregulated in patients with obstructive nephropathy. The expression of GNMT in the renal interstitium of kidney tissues was determined by immunohistochemistry. The data are presented as means ± standard errors. * *p* < 0.05 versus patients without obstructive nephropathy.

**Figure 2 ijms-24-06859-f002:**
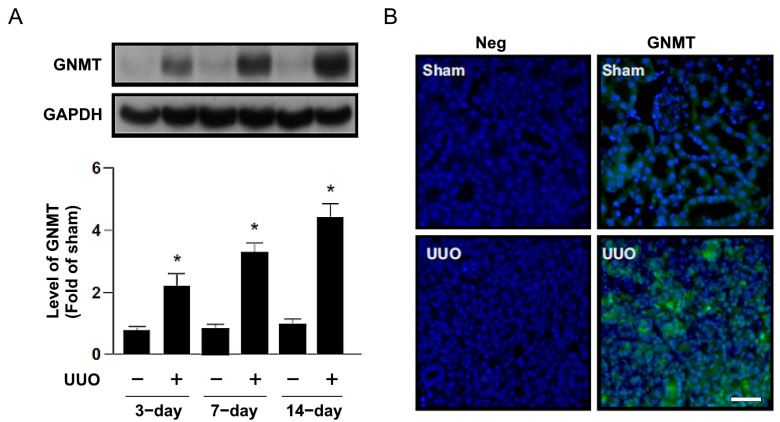
UUO upregulated GNMT expression in wild-type mice. (**A**) GNMT level in WT mice 3, 7 and 14 days after undergoing UUO surgery. The protein level of GNMT was analyzed by Western blotting. GAPDH was used as a loading control. (**B**) WT mice were observed for 14 days after undergoing UUO surgery. The protein expression of GNMT was measured by immunofluorescence. Neg: negative control. Magnification: 200×. Scale bar: 50 μm. The data are presented as means ± standard errors. * *p* < 0.05 versus the sham group.

**Figure 3 ijms-24-06859-f003:**
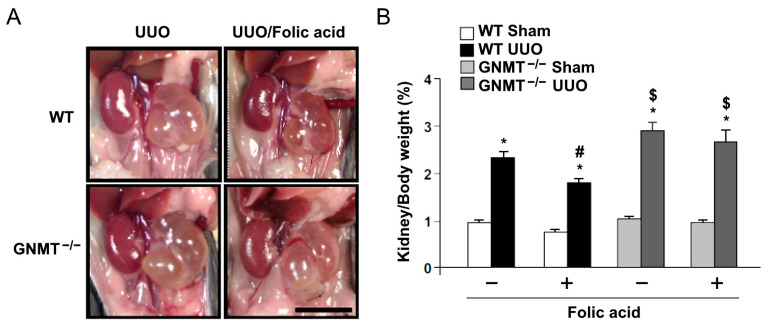
Effect of folic acid on UUO-induced hydronephrosis in WT and *GNMT^−/−^* mice. (**A**) WT and *GNMT^−/−^* mice underwent UUO and received folic acid for 14 days. At the end of the experiment, mice were euthanized by CO2 and the kidneys were then photographed. (**B**) Both sham and UUO-treated kidneys were harvested and weighed. The development of hydronephrosis was evaluated via normalization to body weight. The data are presented as means ± standard errors. * *p* < 0.05 versus the sham group, ^#^ *p* < 0.05 versus non-folic acid-treated mice, ^$^ *p* < 0.05 versus WT mice.

**Figure 4 ijms-24-06859-f004:**
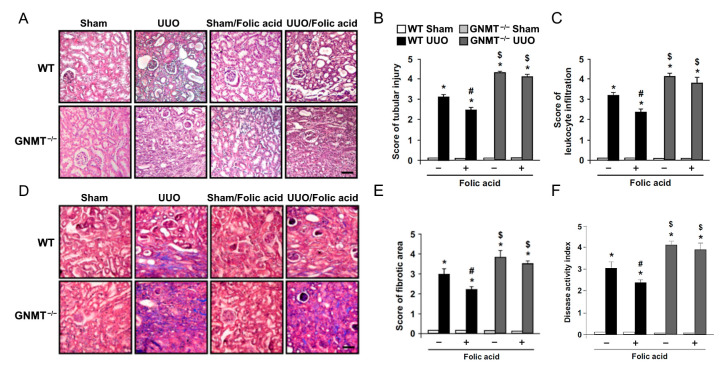
Effect of folic acid on UUO-induced renal tubular injury, leukocyte infiltration, and interstitial fibrosis in WT and *GNMT^−/−^* mice. Paraffin-embedded kidney specimens were cut into 8 µm sections for histological examination. Magnification: 200×. Scale bar: 50 µm. (**A**) Representative hematoxylin and eosin staining of renal tubular injury and leukocyte infiltration of kidney tissue. Both the degree of (**B**) renal tubular injury and (**C**) leukocyte infiltration were recorded. (**D**) Masson’s trichrome staining of collagen deposition in kidney tissue. (**E**) The degree of fibrotic area was recorded. (**F**) The average tubular injury, leukocyte infiltration, and fibrotic area scores were used as disease activity indices to indicate the degree of renal injury. The data are presented as means ± standard errors. * *p* < 0.05 versus the sham group, ^#^ *p* < 0.05 versus non-folic acid-treated mice, ^$^ *p* < 0.05 versus WT mice.

**Figure 5 ijms-24-06859-f005:**
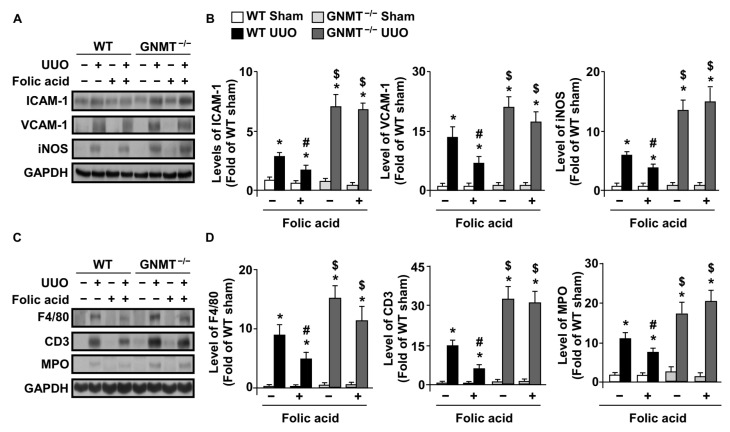
Effect of folic acid on UUO-induced renal inflammation and leukocyte infiltration in WT and *GNMT^−/−^* mice. (**A**,**B**) Tissue extracts from kidneys were analyzed via immunoblotting with antibodies against the adhesion molecules ICAM-1 and VCAM-1, as well as inflammatory markers of iNOS. (**C**,**D**) Tissue extracts from kidneys were also analyzed via immunoblotting with antibodies against leukocyte markers, including F4/80, CD3, and MPO. * *p* < 0.05 versus the sham group, ^#^ *p* < 0.05 versus non-folic acid-treated mice, ^$^ *p* < 0.05 versus WT mice.

**Figure 6 ijms-24-06859-f006:**
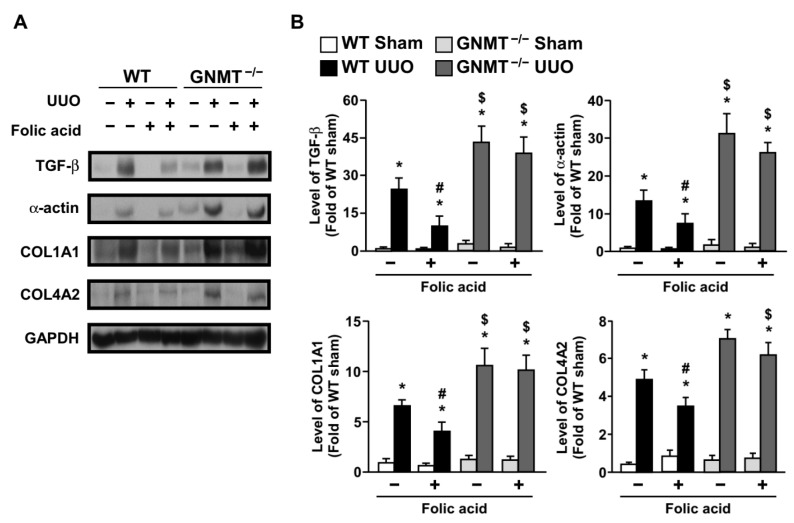
Effect of folic acid on UUO-induced expression of fibrosis-related proteins in WT and *GNMT^−/−^* mice. (**A**,**B**) Tissue extracts from kidneys were analyzed by immunoblotting with antibodies against the fibrosis-related proteins TGF-β, α-SMA, COL1A1, and COL4A2; GAPDH was used as a loading control. * *p* < 0.05 versus the sham group, ^#^ *p* < 0.05 versus non-folic acid-treated mice, ^$^ *p* < 0.05 versus WT mice.

**Figure 7 ijms-24-06859-f007:**
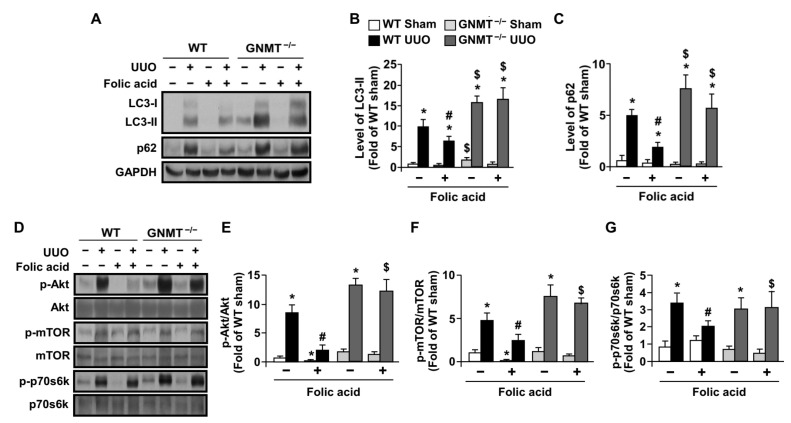
Effect of folic acid on UUO-induced autophagy and its regulatory pathway in WT and *GNMT^−/−^* mice. (**A**–**C**) Tissues extracted from kidneys were analyzed via immunoblotting with antibodies against the autophagy markers LC-3 and p62; GAPDH was used as a loading control. (**D**–**G**) Tissues extracted from kidneys were analyzed via immunoblotting with antibodies against autophagy-related signaling pathway components: these included phosphorylated Akt, mTOR, and p70s6k, as well as total Akt, mTOR, and p70s6k. * *p* < 0.05 versus the sham group of WT mice, ^#^ *p* < 0.05 versus non-folic acid-treated UUO WT mice, ^$^ *p* < 0.05 versus folic acid-treated UUO WT mice.

**Figure 8 ijms-24-06859-f008:**
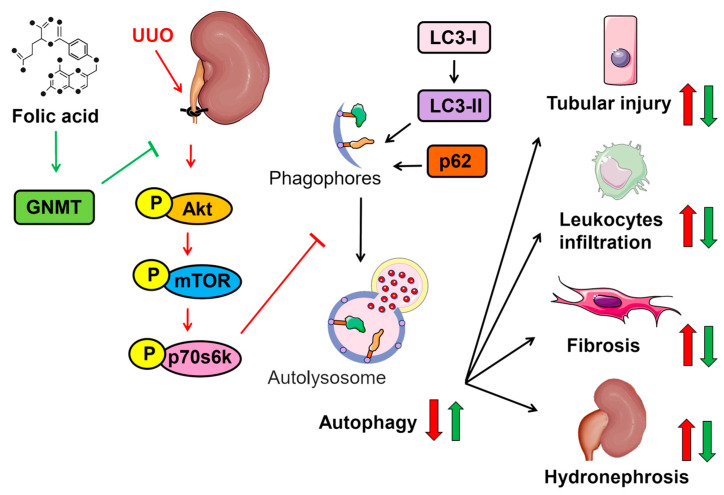
Schematic representation of the protective effect of folic acid on experimental obstructive nephropathy. UUO-induced obstructive nephropathy leads to the development of renal tubular injury, leukocyte infiltration, interstitial fibrosis, and hydronephrosis via an increase in autophagy. Administration of folic acid reduces UUO-induced renal injury by decreasing the level of autophagy in a GNMT-dependent manner.

**Table 1 ijms-24-06859-t001:** Microscopy assessment of histological changes in UUO-induced renal injury.

	0	1	2	3	4	5
Tubular injury	Normal	<10%	10~25%	25~50%	50~75%	>75%
Leukocyte infiltration	Normal	<10%	10~25%	25~50%	50~75%	>75%
Fibrotic area	Normal	<10%	10~25%	25~50%	50~75%	>75%

Disease activity index = (Tubular injury + Leukocyte infiltration + Fibrotic area)/3.

## Data Availability

The datasets used and/or analyzed during the current study are available from the corresponding author on reasonable request.

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
