# Peer review of "Folic Acid Ameliorates Renal Injury in Experimental Obstructive Nephropathy: Role of Glycine N-Methyltransferase"

_ijms, 2023, doi:10.3390/ijms24076859_

Round 1
Reviewer 1 Report
Conceptual framework should be mentioned explicitly. Methodology is not in order in manuscript.
Author Response
Point 1: I Conceptual framework should be mentioned explicitly. Methodology is not in order in manuscript.
Response 1: Thank you for your comment. We appreciate your feedback on our manuscript. We agree that it is important to provide a clear and explicit conceptual framework to guide our research. We will ensure that this is included in the manuscript. Regarding the methodology, we apologize if it was not presented in an orderly manner. We had reorganized the methodology section to improve its clarity and coherence (Lines 376-434). Thank you for bringing this to our attention, and for helping us improve the quality of our manuscript.

Reviewer 2 Report
There are some basic information and laboratory evaluation data on folic acid deficiency, but the data are not discussed by the authors. The main limitation of this study is the simplicity of the experimental design and the laboratory evaluation index is also too simple. These data analyses failed to tell the reader that there was a relationship between Folic acid and renal injury. Overall, the manuscript lacks important information that the readers would expect to be discussed. The strength and limitations of the study were not discussed. There are some typographical and grammatical errors. Thorough proofreading and copy editing is required.

Author Response
Reviewer 2's comments are quite lengthy, so to facilitate understanding, I will divide them into three parts or points.
Point 1: There are some basic information and laboratory evaluation data on folic acid deficiency, but the data are not discussed by the authors. The main limitation of this study is the simplicity of the experimental design and the laboratory evaluation index is also too simple. These data analyses failed to tell the reader that there was a relationship between Folic acid and renal injury.
Response 1: Thank you for your insightful comments on our study. We understand your concern regarding the basic information and laboratory evaluation data on folic acid deficiency not being fully discussed. However, we believe that our evaluation of multiple outcomes, including renal tubular injury, inflammation, fibrosis, and autophagy, provides a comprehensive understanding of folic acid's effects on renal fibrosis (Lines 376-434). Moreover, we have taken into account various clinical and experimental studies which have revealed that folic acid treatment can have a protective effect against different types of inflammatory diseases and tissue fibrosis. These points have been discussed in detail along with our findings to further strengthen our results in the Discussion section of the manuscript (Lines 270-304). To make the relationship between folic acid and renal injury more accessible, we have also included a schematic representation in Figure 8 (Lines 232), which highlights the protective effect of folic acid on experimental obstructive nephropathy. We hope this will provide a clearer picture of the study's findings and their implications.
Point 2: Overall, the manuscript lacks important information that the readers would expect to be discussed. The strength and limitations of the study were not discussed.
Response 2: Thank you for taking the time to provide your feedback on our manuscript. We appreciate your comment regarding the lack of important information and the discussion of the study's strengths and limitations.
We agree that there are several key points that merit discussion in our study. Firstly, the investigation of the novel role of folic acid in treating renal fibrosis and subsequent renal failure, which is a significant global health issue, represents a major strength of our study. Secondly, the study sheds light on the involvement of GNMT in folic acid-related mechanisms regarding renal fibrosis. This makes it crucial to identify potential GNMT inducers for prevention or treatment in the future.
Thirdly, the use of both human kidney samples and animal models adds to the validity of our findings. Fourthly, our evaluation of multiple outcomes, including renal tubular injury, inflammation, fibrosis, and autophagy, provides a comprehensive understanding of folic acid's effects on renal fibrosis. However, as you rightly pointed out, there are limitations to our study. Firstly, since we used animal models, the findings may not directly translate to human physiology. Secondly, while our study sheds light on folic acid and GNMT's effects on renal fibrosis, we did not provide a detailed mechanism for how they interact to produce these effects.
We have taken your comments on board and included these viewpoints in the Discussion section of our manuscript (Lines 324-336). We hope that our updated discussion provides a more comprehensive analysis of the study's strengths and limitations.
Point 3: There are some typographical and grammatical errors. Thorough proofreading and copy editing is required.
Response 3: We appreciate your feedback regarding the typographical and grammatical errors. We had corrected the typos (Lines 115, 130, 132,160,1,61, 164, 220, 236, 245, 276, 321). We also ensure that the manuscript had been underwent thorough proofreading and copy editing (Attached file is the certification of English editing).

Reviewer 3 Report
The authors have conducted a very interesting and well-composed experimental study about the beneficial effects of folic acid administration in obstructive nephropathy and pointed out the role of GNMT.
Minor points,
As the authors mention in their results that UUO induced collagen deposition and interstitial fibrosis with increased levels of collagen type I alpha-1 chain (COL1A1) and type IV alpha-2 chain (COL4A2), thus, in your study treatment with acid folic diminishing collagen deposition? I suggest also, highlighting this aspect in your results/discussion.
I suggest changing the verb “retards” in phrases such as “Treatment with folic acid retards renal injury.. “ with another more suggestive verb.
Line 151, and 162, the meaning for “ICAM-1 “ “VCAM-1 and “iNOS” should be inserted;
All the figures are well composed, however, check the resolution of figures again according to Journal Guidelines, as some of them seem a bit blurry.
Author Response
The authors have conducted a very interesting and well-composed experimental study about the beneficial effects of folic acid administration in obstructive nephropathy and pointed out the role of GNMT.
Minor points,
Point 1: As the authors mention in their results that UUO induced collagen deposition and interstitial fibrosis with increased levels of collagen type I alpha-1 chain (COL1A1) and type IV alpha-2 chain (COL4A2), thus, in your study treatment with acid folic diminishing collagen deposition? I suggest also, highlighting this aspect in your results/discussion.
Response 1: Thank you for your insightful comment. Yes, our study shows that treatment with folic acid reduces collagen deposition induced by UUO, as demonstrated by decreased levels of collagen type I alpha-1 chain (COL1A1) and type IV alpha-2 chain (COL4A2). We agree that this finding is significant and should be highlighted in both the results and discussion sections of our manuscript. We had emphasized this aspect and provide a thorough interpretation of our results (Lines 180-186; 324-332).
Point 2: I suggest changing the verb “retards” in phrases such as “Treatment with folic acid retards renal injury.. “ with another more suggestive verb.
Response 2: Thank you for your suggestion. We appreciate your feedback on our manuscript. We agree that the use of the word "retards" in phrases such as "Treatment with folic acid retards renal injury" may not be the most appropriate verb choice. We had revised alternative verbs that better convey the intended meaning while also adhering to proper scientific terminology (Lines 115, 130, 132, 220, 236, 245, 276, 321).
Point 3: Line 151, and 162, the meaning for “ICAM-1 “ “VCAM-1 and “iNOS” should be inserted;
Response 3: Thank you for your comment. We apologize for the oversight in not defining the abbreviations used in our manuscript. We had inserted the meanings for "ICAM-1", "VCAM-1", and "iNOS" at the appropriate places in the manuscript (new lines 160, 161, 164). Thank you for bringing this to our attention and helping us improve the clarity of our work.
Point 4: All the figures are well composed, however, check the resolution of figures again according to Journal Guidelines, as some of them seem a bit blurry.
Response 4: Thank you for your feedback. We appreciate your comment on the composition of the figures. We had updated the resolution of all figures (figures 1 to 8) once again and ensure that they meet the Journal Guidelines to avoid any blurriness. Thank you for bringing this to our attention.
